



# Technical note: comparison of water vapor sampling techniques for stable isotope analysis

César Dionisio Jiménez–Rodríguez[1,2], Miriam Coenders–Gerrits[1], Thom Bogaard[1], Erika Vatiero[1,3], and Hubert Savenije[1]

[1]Delft University of Technology. Water Resources Section. Stevinweg 1, 2628 CN Delft, The Netherlands.
[2]Tecnológico de Costa Rica. Escuela de Ingeniería Forestal. 159-7050, Cartago, Costa Rica.
[3]Università degli Studi della Campania Luigi Vanvitelli. Department of Environmental, Biological and Pharmaceutical Sciences and Technologies. Via Vivaldi, 43-81100. Caserta, Italy.

**Correspondence:** César Dionisio Jiménez–Rodríguez (cdjimenezcr@gmail.com)

**Abstract.** Water vapor samples are key elements to describe the evaporation process thanks to the stable isotope signatures of $\delta^2$H and $\delta^{18}$O. However, its sampling is a difficult task that can introduce errors due to isotopic fractionation. This study investigates the consistency of different sampling techniques for atmospheric water vapor. The isotope signature of a parcel of air was determined with a cavity output spectroscopy device during a period of 3 hours (benchmark). This parcel of air was sampled

simultaneously with 3 types of sampling bags made of different materials (metalized polyethylene -MPE-, polyvinyl fluoride -PVF-, low density polyethylene -LDPE-) and with 2 cryogenic baths running at two different pumping rates ($3\,\mathrm{L\,min^{-1}}$ and $50\,\mathrm{mL\,min^{-1}}$). The tested water vapor sampling techniques differ in their ability to keep reliable measurements after sampling and are highly susceptible to procedural errors. MPE bags are the best option for measuring samples up to two weeks of storage after sampling. LDPE and PVF bags are only reliable if the measurement is performed on the same sampling day.

**1  Introduction**

The use of stable water isotope signatures as tracers is a well known practice in environmental sciences (Fry, 2006; Silvertown et al., 2015; Nielsen et al., 2018). Stable water isotopes ($\delta^{18}$O and $\delta^2$H) have been widely used to track different processes such as precipitation (Allen et al., 2016; Ingraham, 1998), percolation (Barnes and Turner, 1998), runoff (Gou et al., 2018; Song et al., 2017), plant water use (Dawson and Ehleringer, 1998; West et al., 2006) and soil evaporation (Barnes and Turner,

1998; Xiao et al., 2018). Sampling techniques for precipitation, soil water or river runoff only require the direct collection of liquid samples. However, sampling procedures to determine plant water use or evaporated water often involve vapor samples. These water vapor samples are good descriptors of the evaporation process, which is considered the second largest flux in the hydrological cycle (Coenders-Gerrits et al., 2014; Miralles et al., 2011; Wang et al., 2014). It is formed by water vapor originated from plant transpiration, soil evaporation and the evaporation of intercepted water on wet surfaces (Savenije, 2004).

Many studies aim to determine the source of the water vapor (Wen et al., 2016; Xiao et al., 2018). This partitioning of evaporation has been carried out with hydrometric data supported with water stable isotopes (Blyth and Harding, 2011; Dubbert et al., 2017; Lawrence et al., 2007; Silvertown et al., 2015; Wang and Yakir, 2000). However, the sampling of water vapor has been





a difficult task since some methods (e.g., cryogenic extraction, cryogenic bath) involve a physical change (Fischer et al., 2019; IAEA, 2016; Orlowski et al., 2016).

Currently, the evaluation of the isotope signatures of water vapor can be performed with three different methods. Firstly,
the Craig-Gordon model (CG–model) (Craig and Gordon, 1965) was developed to determine the water vapor signature of evaporation originated from open waters (Horita et al., 2008) and has also been applied in transpiration and soil evaporation studies (Dubbert et al., 2013; Ferrio et al., 2009; Williams et al., 2004). The CG–model includes the equilibrium effect (as a consequence of the change from liquid to vapor) as well as the kinetic effect (due to the diffusion of heavy and light isotopes of water vapor in the air) (Roden and Ehleringer, 1999). The model is based on the differentiation of three interface layers
between the liquid water and atmosphere. The liquid boundary layer is dominated by the equilibrium fractionation process, followed by a layer dominated by the molecular diffusivity and the last layer is characterized by the turbulent mixing (Craig and Gordon, 1965; Gat, 2008; Horita et al., 2008; Roden and Ehleringer, 1999). The high sensitivity of $\delta^{18}$O to temperature makes some assumptions of this model unreliable for the application in soil evaporation or plant transpiration processes (Dubbert et al., 2013). Thus can lead to important differences ($\sim 10\%o$) in $\delta^2$H and $\delta^{18}$O signatures between the modelled and measured
signatures (Horita et al., 2008).

The second method recollects liquid samples from vapor with cryogenic baths, using liquid nitrogen or the combination of ethanol and dry ice as cooling agents (IAEA, 2016; Kool et al., 2014). This method requires freezing water vapor conveyed with a constant air flow into the collection canister submerged in the cooling agent (He and Smith; Sheppard, 1958; Wen et al.,
2016). A reliable sample should collect 100% of the water vapor, otherwise heavier isotope signatures in the collected sample will occur (Griffis, 2013), which can lead to false conclusions. The consequences of incomplete sample recoveries are similar to the effects as studied by Orlowski et al. (2018) for soil water extraction, where incomplete recoveries lead to fractionated samples.

The most recent method involves direct measurements of stable isotopes in water vapor using mass spectrometers or cavity output spectroscopy. This method has been carried out in Arctic conditions (Steen-Larsen et al., 2013; Steen-Larsen et al., 2015), oceanic climates (Steen-Larsen et al., 2014) and croplands (Wen et al., 2016). The deployment of these devices in the field have a high demand of infrastructure such as a cabin with controlled room temperature and a constant power supply. These conditions exerts pressure on projects with reduced budgets or with remote access. For these cases a way of sampling
and storing water vapor in the field to later analyze them in the laboratory is needed. However, to the authors best knowledge, no sample storage unit has been tested for their applicability to quickly store water vapor in the field and analyze the storage units with a mass or cavity spectrometer in the laboratory.

While this storage unit does not exist for air vapor, some authors have been able to analyze the stable isotopes of water vapor
from small volumes ($< 1.0$ L) to determine the isotope signature of soil water samples under equilibrium (Gralher et al., 2018;





Hendry et al., 2015; Herbstritt et al., 2014; Wassenaar et al., 2008). They underline the risk of using different sampling materials during the storage of soil samples, as a result of water diffusion through the container wall. If their findings for soil samples hold for water vapor as well is still unknown. The aim of this work is to evaluate three types of storage units to determine their suitability for sampling, storing and analyzing water vapor isotopes.

## 2   Methodology

### 2.1   Instrumentation and Measurements

A dual phase Water Isotope Analyzer (WIA; model 912) from Los Gatos Research (LGR) was used to determine the isotope signature of the water vapor and liquid samples. The Water Vapor Isotope Analyzer (WVIA) setup connects the WIA to the

LGR Water Vapor Isotope Standard Source (WVISS; model 908-0004-9002). This was used to provide a controllable flow of water vapor with a known liquid standard measurement for an absolute calibration of the raw measurements of the signatures of both stable isotopes: $\delta^2$H and $\delta^{18}$O. The WVISS was set to run the automatic pump with the following voltages 3.0 V, 2.0 V, 1.5 V and 1.0 V to provide a controlled water vapor concentration (ppm) during the calibration of each set of samples. The dry air needed for the WVISS was provided by the Dry Air Source (DAS) device from LGR. The device pumping rate for all the

samples was fixed at $\sim 90\,\mathrm{mL\,min^{-1}}$. The WIA and the WVISS were attached to a Multiport Inlet Unit (MIU; model: LGR 908-0003-9002) for the automatic control of eight inlets to measure multiple samples for specific periods of time (Figure 1). The MIU has eight ports for 6 mm diameter tubing which allows the development and attachment of different sampling devices. In all the measurements, the first MIU inlet was attached to an altered air source. This altered air source had a distinctive isotope signature compared to the samples and was used to identify among the different samples of other MIU inlets during the

post-processing of the data. The altered air source was achieved by conveying laboratory air through a 2 L borosilicate bottle that was filled with 1.5 kg of silica gel to modified the laboratory air signature to a concentration lower than 4000 ppm.

All vapor samples were measured for 5 min with the WVIA, with sampling intervals of 5 s. The first 3 minutes include the dead volume of the pipeline (50 cm) and the memory effect from the previous sample. The last 2 minutes were used to calculate

the isotope signature of each air sample, obtaining the average and standard deviation of each measurement.

Liquid samples (see Section 2.3) were measured with the Liquid Water Isotope Analyzer (LWIA, see "Liquid Mode" in Figure 1). This setup connects the WIA with a liquid autosampler, injecting $900\,\mu\mathrm{L}$ into a heating chamber for complete vaporization of the water and transferred into the WIA for its measurement. The correction and calibration of the isotope sig-

natures of the liquid samples were performed with the software Laboratory Information Management System (LIMS, version 10.083) for Light Stable Isotopes, version 10.083 for Microsoft Access 2007–2013 (Coplen, 2000).





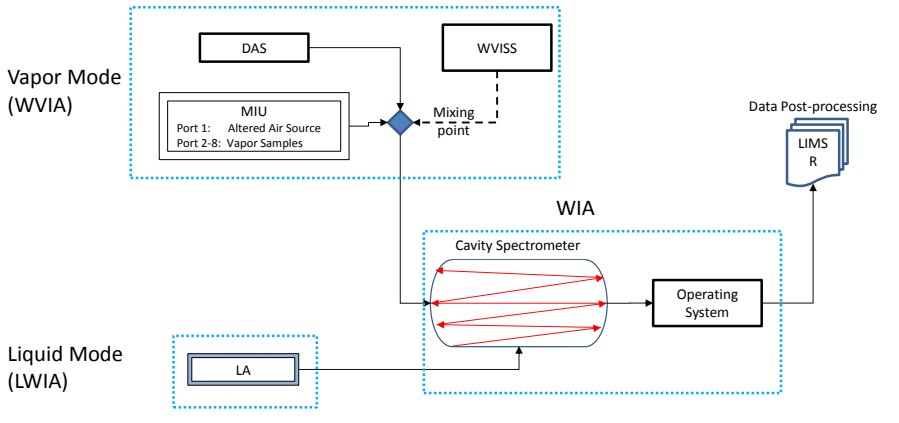

**Figure 1.** Setup of the Water Isotope Analyzer (WIA) used in this experiment. The selection between liquid and vapor mode depends on the type of samples to be analyzed.

Isotope signatures ($^2$H and $^{18}$O) of both sample types (liquid and gas) were expressed in respect to the Vienna Standard Mean Ocean Water (VSMOW) following equation 1 (Craig, 1961). In this equation $\delta$ is the relative concentration (‰) of the stable isotope $^2$H or $^{18}$O, $R$ is the stable isotope ratio ($^2$H/$^1$H or $^{18}$O/$^{16}$O) of the standard water ($R_{\mathrm{standard}}$) and the sample ($R_{\mathrm{sample}}$).

$$\delta = \left( \frac{R_{\mathrm{sample}}}{R_{\mathrm{standard}}} - 1 \right) ‰ \tag{1}$$

## 2.2 Water Vapor Correction

The measurements of the water vapor depends on the concentration of water molecules (ppm) and the specific drift of the laser spectrometer of the WIA unit, which makes it essential to correct each individual measurement (Aemisegger et al., 2012;
10 Rambo et al., 2011; Kurita et al., 2012; Steen-Larsen et al., 2013, 2014). The correction of water vapor measurements was performed with the injection of a standard water with a known isotopic signature at different ppm concentrations. The injection is controlled by a built–in software package that managed the WVISS pump and the DAS. This system allows the use of only one standard water to correct the isotope signature measurements carried out with the WVIA.



The correction was performed with three selected water molecule concentrations depending on the air sample concentrations. A calibration run was performed before every time the MIU start a new set of measurements for both, along the sampling time and during the measurement of the collected samples. The raw signatures of $\delta^2$H and $\delta^{18}$O are calibrated using the correction factors ($\phi_O$ and $\phi_H$) determined based on the dependency of raw signatures to the water mixing ratio ($w$) in ppm. The polynomial coefficients $a$, $b$ and $c$ in the equations 2 and 3 were determined for every set of measurements per experiment (Rambo et al., 2011; Kurita et al., 2012; Steen-Larsen et al., 2013, 2014). The corrected values of each stable isotope ($\delta^{18}$O and $\delta^2$H) were determined with equations 4 and 5, where the $\delta^{18}$O$_{\text{raw}}$ and $\delta^2$H$_{\text{raw}}$ are the raw measurements given by the device (Rambo et al., 2011; Steen-Larsen et al., 2013).

$$\phi_O = \frac{\delta^{18}O_{\text{raw}}}{\delta^{18}O_{\text{standard}}} = a_O w^2 + b_O w + c_O \tag{2}$$

$$\phi_H = \frac{\delta^2 H_{\text{raw}}}{\delta^2 H_{\text{standard}}} = a_H w^2 + b_H w + c_H \tag{3}$$

$$\delta^{18}O = \frac{1}{\phi_O} \delta^{18}O_{\text{raw}} \tag{4}$$

$$\delta^2 H = \frac{1}{\phi_H} \delta^2 H_{\text{raw}} \tag{5}$$

### 2.3 Experimental Design

This experiment tested whether the stored mass of vapor remained unchanged as well as whether the isotope signature of the stored air samples remain consistent in time. The 3 types of bags selected to store air samples were (see Appendix A):

1. MPE: these 1 L bags of methalized polyethylene were manufactured with a five layer structure, with an aluminium layer in between (Commercial name: Foil Bag). According with the supplier, the sample stability is 5 days for low molecular compounds such as $CH_4$, $CO_2$, CO, and other non specified gases.

2. PVF: these 1 L bags were made of polyvinyl fluoride (Commercial name: Tedlar Bag).

3. LDPE: these 1.1 L bags were made of low density polyethylene used for filling packaging spaces.

Both, MPE and PVF have been designed to be filled to 90 % of its volume capacity and every bag has a 2-in-1 polytretrafluorethylene (PTFE) fitting for the injection and extraction of the air sample. Whilst the LDPE are fabricated with a simple valve made from polyethylene as well, a special inlet was built to connect the LDPE to the individual ports of the MIU (see Appendix





B).

During a period of 3 hours, 18 air samples were collected per bag type (6 samples per hour) in the laboratory. Only 15 samples per bag type were used for the analysis, leaving 3 samples as a backup for replacements if needed. Air samples were
collected manually with a medical cardiopulmonary resuscitation (CPR) balloon with a conical plastic inlet that allows to push the air into the sample bags. Also, the WVIA was analyzing the laboratory air during the sample collection. This isotope signature was used as Benchmark for the analysis. Simultaneously, two sets of liquid samples were collected with a cryogenic bath at two different pumping rates ($3\,\mathrm{L\,min^{-1}}$ and $50\,\mathrm{mL\,min^{-1}}$). The cryogenic baths were built with a test tube of $50\,\mathrm{mL}$ capacity immerse in a container filled with ethanol (100\%) inside a cooler filled with dry ice (-70\,°C). The water collected in
both test tubes was thaw and transferred to a $1.5\,\mathrm{mL}$ vial for its measurement after the experiment with the LWIA.

The isotope signature of water vapor was measured from the stored air samples during a period of 16 days. The air was analyzed on the sampling day ($T_0$) and after 1, 2, 9, and 16 days after collection ($T_1$, $T_2$, $T_9$, $T_{16}$, respectively). All the air samples were compared against the Benchmark (i.e, the laboratory air sampled on $T_0$ with the WVIA).

## 2.4   Analysis

The consistency analysis of the isotopic signatures was performed comparing the isotope signatures against the Benchmark. The cross comparison was performed with the Z-score analysis (Equation 6) (Orlowski et al., 2016; Wassenaar et al., 2012). Where $S$ is the isotope signature ($\delta^2$H or $\delta^{18}$O) of the bags and cryogenic samples, $B$ is the benchmark isotope signature
(WVIA) and $\mu$ is the target variability. Differing from Orlowski et al. (2016) and Wassenaar et al. (2012), $\mu$ was established as the isotope range measured with the WVIA during the 3 hours of measurements of the benchmark ($\delta^2$H: $2.0\,‰$ and $\delta^{18}$: $0.4\,‰$) considering the transient condition in the laboratory. Thus we adopted the limits proposed by Orlowski et al. (2016) for classifying the samples as accurate ($Z_{\mathrm{score}} < 2.0$), questionable ($Z_{\mathrm{score}}$: 2.0–5.0) or unacceptable ($Z_{\mathrm{score}} > 5.0$).

$$Z_{\mathrm{score}} = \frac{S - B}{\mu} \tag{6}$$

After the initial analysis, it was necessary to test the ability of the cryogenic bath to fully collect all air water vapor. Consequently, the cryogenic bath with a pump rate of $3\,\mathrm{L\,min^{-1}}$ was tested a second time in duplicate for 4 hours. The isotope signatures of the cryogenic bath samples were compared against the WVIA water vapor signatures collected during the additional sampling. The comparison was performed with the $Z_{\mathrm{score}}$ analysis. All data processing and analysis were performed with the software R (R Core Team, 2017).






## 3   Results and Discussion

In Figure 2[A] the results of the isotopic consistency are depicted. Here the isotope signature of the benchmark during the 3 hours experiment had an isotope signature of -15.61±0.14‰ and -115.12 ± 0.47‰ for $\delta^{18}$O and $\delta^{2}$H, respectively. All the vapor samples collected with the bags that were measured on the same sampling day are classified as accurate samples

based on the $Z_{\text{score}}$ (Figure 2 [A]). The signature of the laboratory air changed during every set of measurements since the sampling was performed. The orange samples marked as "Laboratory" in Figure 2 [A] depict isotopic signatures of the air of the laboratory during analysis. These differences in laboratory air signatures influences the measurement results from all the air samples collected with the three types of bags but not all to the same degree. The MPE samples are the only bags used in this experiment with almost all measurements located within the accurate region of the $Z_{\text{score}}$ plot ($Z_{\text{score}} < 2$). Despite the accuracy

provided with the MPE, the measurements are influenced by the isotope signature of the air within the laboratory. All the measurements after the sampling date with the LDPE and PVF bags are located within the questionable region of the $Z_{\text{score}}$ plot ($Z_{\text{score}}$: 2-5), while the PVF samples from $T_9$ are in the unacceptable region ($Z_{\text{score}} > 5$). These sampling bags are influenced by the isotopic signature of the laboratory air considering its location close to the laboratory signature during the measurements.

Contrary to the samples collected with the sampling bags, the liquid samples from the cryogenic baths were tagged as unacceptable ($Z_{\text{score}}$ values $> 5$, see Figure 2[B]). The cryogenic sample with pumping rate of $50\,\text{mL min}^{-1}$ is only based on one sample because it was only possible to collect a sample of 0.1 mL during the 3 hours of sampling, while the $3\,\text{L min}^{-1}$ collected 0.25 mL every hour. Both liquid samples show a heavier signature than the benchmark during the first sampling, denoting an incomplete condensation of the water vapor in the lab. These differences are likely linked to a not perfect collection efficiency

during the cryogenic sampling with the cold traps (Griffis, 2013). Aiming to discard wrong assumptions about the cryogenic bath, the second sampling after the analysis (Figure 2 [B]) showed a better proficiency of the sampling procedure with the $3\,\text{L min}^{-1}$ pumping rate and the cryogenic bath. During this sampling, only 3 samples were tagged as questionable samples while their duplicates were tagged as accurate samples. These inconsistencies among the quality of liquid samples collected with cryogenic baths depends on the capacity to collect all the water vapor from the conveyed air.


The Water Vapor Transmission Rate (WVTR) of each material provides insights about the reason behind the variation in the stable isotope measurements, including the MPE bags. Thus the WVTR defines the ability of a film to transfer water molecules depending on the relative humidity gradient (Kumaran, 1998; Keller and Kouzes, 2017). Note that the diffusion characteristics of foil layered materials are directly influenced by temperature and air vapor concentration (Pons et al., 2014). Among the 3

types of sampling bags in this experiment, the MPE bags have the lowest WVTR reported ($0.09\,\text{g m}^{-2}\text{d}^{-1}$), followed by the LDPE bags between $0.39$–$0.59\,\text{g m}^{-2}\text{d}^{-1}$ and PVF with the highest value of $0.83\,\text{g m}^{-2}\text{d}^{-1}$ (Keller and Kouzes, 2017; Tock, 1983).



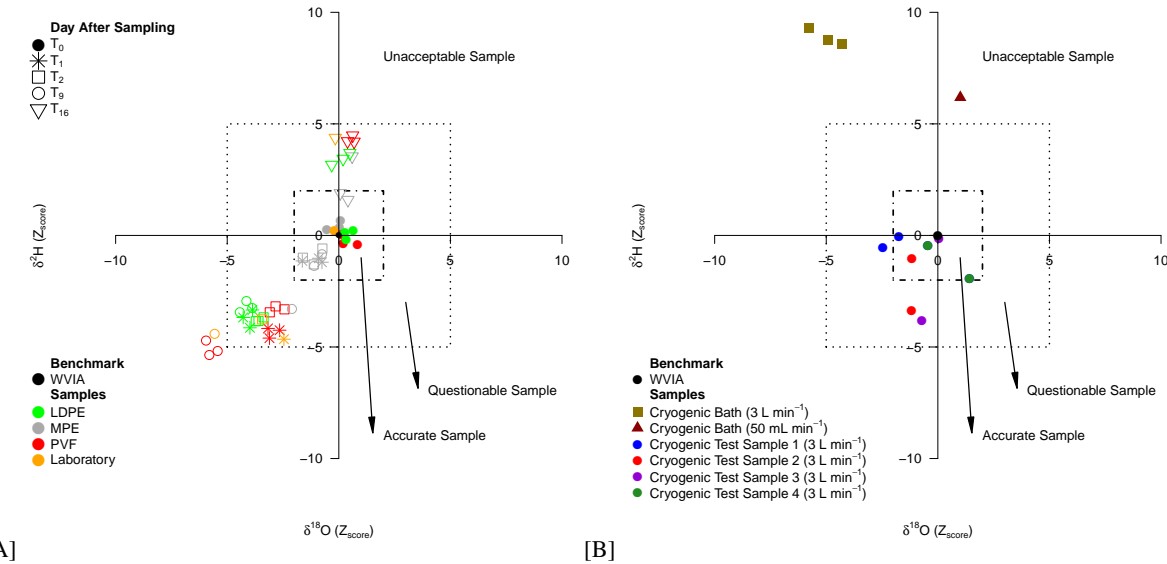

**Figure 2.** Dual plot for the $Z_{score}$ of $\delta^2H$ and $\delta^{18}O$ of vapor samples. [A] is the main analysis of the sampling bags and [B] is the additional cryogenic bath test performed.

## 4 Conclusions

This study investigates the consistency of different sampling techniques to collect atmospheric water vapor for stable isotope analysis. The Low Density Polyethylene (LDPE) and Polyvinyl Fluoride (PVF) bags are influenced by the Water Vapor Transmission Rate (WVTR) of their material. The tested water vapor sampling techniques differ in their ability to keep reli-

5   able measurements after sampling and are highly susceptible to procedural errors. All the sampling bags perform well if the measurements are carried out on the same day of the sampling, keeping $Z_{score}$ values within the accurate zone ($Z_{score} < 2.0$). However, if the samples are required to be stored for longer periods the Methalized Polyethylene (MPE) bags are the best option to obtain reliable signatures after the first day of storage up to two weeks after sample collection. Water vapor sampling with cryogenic baths provides suitable accuracy. However, there is a high risk of incomplete condensation leading to the

10   collection of fractionated water samples.

*Acknowledgements.* This work was carried out with the aid of a grant (863.15.022) from The Netherlands Organization for Scientific Research (NWO) and PINN-MICITT Costa Rica (contract: PED-032-2015-1). Special thanks to H.C. Steen-Larsen for his clarifications about the calibration procedures of laser spectrometers. To Bart Schilperoort for his well-timed comments.



*Competing interests.* The authors declare that they have no conflict of interest.

*Author contributions.* CDJR and MC designed the experiment and performed the data analysis. CDJR and EV collected the data. CDJR prepared the manuscript with contributions from all co-authors.





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



**Appendix A: Sampling bags selected for the experiment**

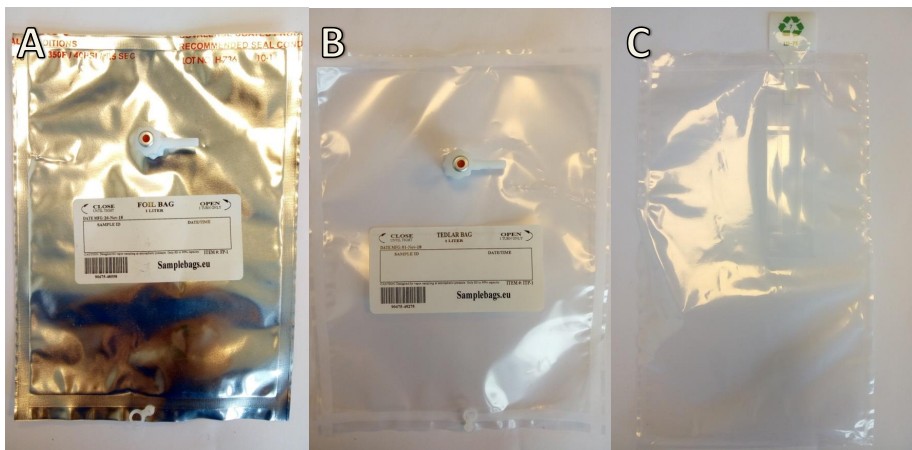

**Figure A1.** Sampling bags selected for the collection and storage of air. A is the methalized polyethylene (MPE) bag, B is the polivinil fluoride (PVF) bag and C is the low density polyethylene (LDPE) bag.

**Appendix B: Connecting inlet for the LDPE bags**

The inlet is composed of four parts allowing to plug in and out each sample bag without mixing the air contained in the bag with the laboratory air (Fig. B1). Parts A, B and D convey the air flow from the sample bag towards the MIU. Part A connects
5  the inlet to the different MIU ports, it has an outer diameter of 6 mm. Part B with a outer diameter of 4 mm and allows for the tight movement of the connector (Part C) along all its length. Part C has a conical shape of 5 mm to 9 mm of outer diameters. This part seals the valve opening preventing to lose air from the sampling bag and allow to adjust to the length of the sampling bag valve. Part D has a conical shape of 1 mm to 5 mm of outer diameters, allowing to insert the full extension of the inlet within the sampling bag valve.





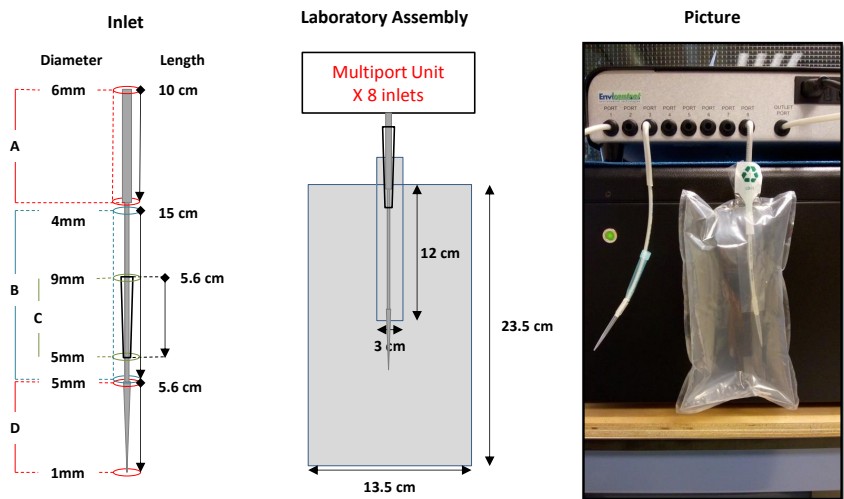

**Figure B1.** Laboratory assembly details of the Multiport Inlet Unit (MIU) connection, sampling bag and the view of the inlet connection with the multiport unit.