# Peer review of "Technical note: comparison of water vapor sampling techniques for stable isotope analysis"

_Hydrology and Earth System Sciences, 2019_

## Referee Comment (RC1) · Anonymous Referee #1 · 26 Aug 2019

With the development of water isotopes measurements using the laser spectrometer, real-time measurements of water vapor isotopes can be realized. However, the real-time measurements of water vapor isotopes need a constant power supply and the deployment of the laser spectrometer in the field is not cheap. Instead, it is more convenient and cheaper to collect air samples in the field for later analysis in the laboratory with a mass or laser spectroscopy. However, the applicability of the sample storage unit needs to be tested. This study airs to test the applicability of different sampling techniques, which can give a guide for scientists who'll collect air samples for isotopic measurements. The paper is well written and has a good readability. It deserves to be published and I have a few suggestions for improving this paper.

[Figure]

Line 20-21 in page 3, why to modify the laboratory air to a concentration lower than 4000 ppm? To my knowledge, the laser spectroscopy generally has lower measurement accuracy for low vapor concentration (for example, <2000 ppm).

Line 1-13 in page 5, how do you establish the relationship (equation 2 or 3) between isotopic ratio and vapor concentration? Because you mentioned in line 12-13 of page 3 'The WVISS was set to run the automatic pump with the following voltages 3.0V, 2.0V, 1.5V and 1.0V to provide a controlled water vapor concentration (ppm) during the calibration of each set of samples.', did you establish the relationship using four data points? If yes, the number of data points for establish the relationship between isotopic ratio and vapor concentration is not enough. The authors should add a plot in the manuscript for manifesting the relationship between isotopic ratio and vapor concentration. In addition, how did you correct the drift effect of the laser spectrometer? which was not mentioned in the text.

In addition to the Water Vapor Transmission Rate (WVTR) of sample bag, other factors such as air tightness of valve or fitting of the sample bag may also have an important influence on the isotopic measurements. Other potential influences should be discussed in the text.

---

## Author Comment (AC1) · 29 Aug 2019

**Reply to anonymous referee #1**

In blue we copied the comments of the reviewer, in black our replies.

With the development of water isotopes measurements using the laser spectrometer, real-time measurements of water vapor isotopes can be realized. However, the real time measurements of water vapor isotopes need a constant power supply and the deployment of the laser spectrometer in the field is not cheap. Instead, it is more convenient and cheaper to collect air samples in the field for later analysis in the laboratory with a mass or laser spectroscopy. However, the applicability of the sample storage unit needs to be tested. This study airs to test the applicability of different sampling techniques, which can give a guide for scientists who'll collect air samples for isotopic measurements. The paper is well written and has a good readability. It deserves to be published and I have a few suggestions for improving this paper.

The authors want to thank the reviewer for his/her suggestions on our manuscript. We welcome the reviewer interest and relevance acknowledgement of the study. The reviewer provided a series of suggestions to improve the manuscript that we want to address point-by-point:

Line 20-21 in page 3, why to modify the laboratory air to a concentration lower than 4000 ppm? To my knowledge, the laser spectroscopy generally has lower measurement accuracy for low vapor concentration (for example, <2000 ppm).

**Reply:**
The idea behind adding an altered air source during the sampling is to be able to differentiate between samples during the laboratory procedure and post-processing of the data. This helped to carry out the data analysis because after the air passed by the drying element, the $\delta^2H$ and $\delta^{18}O$ signatures will change drastically. To clarify this, we proposed to improve the lines 18 and 19 of page 3 as follows:

"… post-processing of the data. **The data obtained from this inlet was not used during the analysis as it was used only as a distinction mark between samples**. The altered air …"

Line 1-13 in page 5, how do you establish the relationship (equation 2 or 3) between isotopic ratio and vapor concentration? Because you mentioned in line 12-13 of page 3 'The WVISS was set to run the automatic pump with the following voltages 3.0V, 2.0V, 1.5V and 1.0V to provide a controlled water vapor concentration (ppm) during the calibration of each set of samples.', did you establish the relationship using four data points? If yes, the number of data points for establish the relationship between isotopic ratio and vapor concentration is not enough.

**Reply:**
This relationship was established based on the correction procedure used during similar experiments and devices published by Rambo et al. (2011), Kurita et al. (2012) and Steen-Larsen et al. (2013, 2014). This is mentioned on line 6 of page 5. Despite it is true that the relationship was established using four data points, each data point corresponds to the average of 12 individual measurements performed every 5 s by the laser. The use of this method to correct the isotope measurements is based on the assumption of a linear drift in the humidity-isotope correction (Steen-Larsen et al., 2013).
We proposed to add on page 3, line 13 the following:

"… of each set of samples, **running each voltage for a period of 2 min**. The dry air needed for the …"

And on page 5, line 6 the following:

"… Steen-Larsen et al., 2013, 2014). **Each data point used in equations 2 and 3 corresponds to the last minute of measurements for each voltage, obtaining an average based on 12 individual measurements for both stable isotopes and water vapor concentrations**. The corrected values of each …"

**Reply:**
The authors agreed with the suggestion of adding an additional plot to the manuscript showing the relationship of the correction procedure applied. This plot will be added as an appendix to the manuscript as follows:

Adding on page 5, line 4 the following:
"… to the water mixing ratio (ω) in ppm **(see Appendix C)**:  "

And the appendix as:

**Appendix C: Plots of the Water Vapor Correction Procedure**

[Figure]

**Figure C1.** Water vapor correction plots showing the variation of the water vapor concentration (ppm) and the correction factors $\phi_O$ and $\phi_H$. Plot A shows the variation of water vapor concentration in ppm for

each voltage used on the WVISS pump during the correction procedure. Plots B and C show the polynomial relationships between the water vapor concentration and the correction factors $\phi_O$ and $\phi_H$, respectively.

In addition, how did you correct the drift effect of the laser spectrometer? which was not mentioned in the text.

**Reply:**

The LGR analyser used during this experiment did not experience a significant drift during the measurements due to the little time running on a daily basis (less than six hours every day). Additionally, the model of water isotope analyser (IWA) keeps a "negligible drift" as it is stated by the manufacturer (LGR, 2019).

To clarify this, we proposed to add the following sentence on Page 4, line 10:
"… Steen-Larsen et al., 2013, 2014). **The drift of the used laser spectrometer was negligible, because the measurement period was not longer than 6 hours every day. In addition, the thermal control within the laser chamber provides stable measurements with a negligible drift as it is stated by the manufacturer (LGR, 2019).** The correction of water vapor measurements was …"

In addition to the Water Vapor Transmission Rate (WVTR) of sample bag, other factors such as air tightness of valve or fitting of the sample bag may also have an important influence on the isotopic measurements. Other potential influences should be discussed in the text.

**Reply:**

The authors followed the suggestion from the reviewer and it was decided to add the following paragraph on page 8, line 1:

"**The tendency of drift towards the signature of the laboratory air could be linked to other factors such as welding quality between bag material and the valve (for MPE and PVF bags), fitting issues between the tubing connecting the sample bags to the MIU unit (all sample bags) or the inlet connection for the LDPE bags. In the case of MPE and PVF bags, the manufacturer states that the bags should not be filled more than 90% of their capacity like we did in the experiment. This practice could lead to the development of fissures between the air valve and the bag material that in the case of PVF bags due to their brittle properties respect to MPE of LDPE bags. An increment on the air pressure within the MPE bags can lead to the detachment of the air valve from the layers in which it is welded. LDPE bags are susceptible to leaking as a consequence of the inlet built with in-house materials that the presence of different joints can induce the filtering of the laboratory air.** "

**References**

LGR: Los Gatos Research, Inc. Isotopic Water Analyzer (Liquid+Vapor) - Enhanced Performance, https://www.lgrinc.com/analyzers/isotope/isotopic-water-analyzer, visited on: August 27, 2019, 2019.

Kurita, N., Newman, B. D., Araguas-Araguas, L. J., and Aggarwal, P.: Evaluation of continuous water vapor δD and δ 18O measurements by off-axis integrated cavity output spectroscopy, Atmospheric Measurement Techniques, 5, 2069–2080, https://doi.org/10.5194/amt-5-2069-2012, 2012.

Rambo, J., Lai, C.-T., Farlin, J., Schroeder, M., and Bible, K.: On-Site Calibration for High Precision Measurements of Water Vapor Isotope Ratios Using Off-Axis Cavity-Enhanced Absorption Spectroscopy, Journal of Atmospheric and Oceanic Technology, 28, 1448–1457, https://doi.org/10.1175/JTECH-D-11-00053.1, 2011.

Steen-Larsen, H. C., Johnsen, S. J., Masson-Delmotte, V., Stenni, B., Risi, C., Sodemann, H., Balslev-Clausen, D., Blunier, T., Dahl-Jensen, D., Ellehøj, M. D., Falourd, S., Grindsted, A., Gkinis, V., Jouzel, J., Popp, T., Sheldon, S., Simonsen, S. B., Sjolte, J., Steffensen, J. P., Sperlich, P., Sveinbjörnsdóttir, A. E., Vinther, B. M., and White, J. W. C.: Continuous monitoring of summer surface water vapor isotopic composition above the Greenland Ice Sheet, Atmospheric Chemistry and Physics, 13, 4815–4828, https://doi.org/10.5194/acp-13-4815-2013, 2013.

Steen-Larsen, H. C., Sveinbjörnsdottir, A. E., Peters, A. J., Masson-Delmotte, V., Guishard, M. P., Hsiao, G., Jouzel, J., Noone, D., Warren, J. K., and White, J. W. C.: Climatic controls on water vapor deuterium excess in the marine boundary layer of the North Atlantic based on 500 days of in situ, continuous measurements, Atmospheric Chemistry and Physics, 14, 7741–7756, https://doi.org/10.5194/acp-14-7741-2014, 2014.

---

## Referee Comment (RC2) · Anonymous Referee #2 · 7 Nov 2019

[referee-annotated manuscript omitted]

---

## Author Comment (AC2) · 12 Nov 2019

**Reply to anonymous referee #2**

In blue we copied the comments of the reviewer, in black our replies.

The study of Jiménez-Rodríguez et al. examines different ways of sampling and storing atmospheric water vapor for water stable isotope analysis in the lab. Their technical note is very well written and the photos and technical drawings provided inthe Supplementary are very helpful and support the description of the experimentsin a very concise way. In addition to questions raised by referee 1, I have a few more questions and suggestions on improving the manuscript that you can find in the uploaded pdf file.

The authors thank the reviewer for his/her additional remarks aiming to improve the manuscript. Following his/her suggestions, we provide a reply for each of them:

Comment 1. Page 1, Line 1: Water vapour samples from which hydrological component?
Reply:  we changed for:

"Atmospheric water vapor samples are key elements …"

Comment 2. Page 1, Line 1: From where? The soil, open water body...? Be more specific.
Reply: we changed for:

"… to describe the different elements of the evaporation process (e.g, plant transpiration, soil evaporation and the evaporation of intercepted water on wet surfaces) thanks to …"

Comment 3. Page 2, Line 28: "has" instead of "have"
Reply: Done.

Comment 4. Page 3, Line 3: add ","
Reply: Done.

Comment 5. Page 3, Line 3: "sampling bag's"
Reply: we changed the sentence changing "storage units" for "sampling bags"

Comment 6. Page 3, Line 4: The "aim of the research" needs a few more sentences.
Reply: we improved the "aim of the research" as follows:

"The aim of this work is to evaluate different sampling procedures to collect atmospheric water vapor and analyze the stable water isotopes. This experiment tested whether the stored mass of vapor remained unchanged as well as whether the isotope signature of the stored air samples remain consistent in time. We included three sampling bags to determine their suitability for sampling, storing and analyzing water vapor isotopes. The results were compared against a set of cryogenic samples and direct measurements performed with a cavity output spectrometer."

Comment 7. Page 3, Line 19: Identify what?
Reply: we used this distinctive isotope signature to identify each of the individual samples. We updated the sentence as follows:

"… compared to the samples. It was used to identify the measurements of each individual sample with the order of the MIU inlets during the post-processing of the data. The altered …"

Comment 8. Page 3, Line 21: "modify" instead of "modified"
Reply: Done.

Comment 9. Page 3, Line 21: add "water vapor"
Reply: Done.

Comment 10. Page 3, Line 25:Did you also test longer measuring times?
Reply: We did not test different measuring times, since after 120 s the isotopic signal did not change.

Comment 11. Page 5, Line 2: Wrong sentence structure.
Reply: We restructure the sentence to be more readable as follows:

"…sample concentrations. This correction was carried out during the water vapor sampling and the posterior measurement of the samples. The correction procedure was performed every time the MIU start a new round of measurements.  The raw signatures …"

Comment 12. Page 5, Line 4: Tenses
Reply: We corrected the tenses as follows: "were" for "are".

Comment 13. Page 5, Line 15-16: This sentence should go at the end of the introduction, too.
Reply: Thanks for the recommendation. We followed your suggestion adding it to improve the aim of the research.

Comment 14. Page 5, Line 18: "to" instead of "with"
Reply: Done.

Comment 15. Page 5, Line 23: Could you please provide the distributor?
Reply: MPE and PVF sampling bags are distributed by "MediSense", while the LDPE bags by "HARDIRON Store".  However, the authors do not think this information is relevant for the manuscript considering that the technical information of the samples is already available (Page 5, lines 17 to 24; and the Appendix A).

Comment 16. Page 6, Line 5: Could you please provide the distributor?
Reply: the CPR balloon is distributed by "Praxisdienst" in The Netherlands but the authors do not think this information is relevant for the manuscript. However, we agree that it requires a bit more of technical information. Consequently, we add the following in page 5, line 6:

"… the sample bags. This device has a balloon with a volume of 1.0 L. It is made with sturdy Polyvinyl Chloride (PVC) with a Positive End-Expiratory Pressure (PEEP) valve to release the air when excess of pressure is present. Also, the WVIA …"

Comment 17. Page 6, Line 9: Could you also provide a technical drawing for this setup?
Reply: We add the following image as an Appendix to the manuscript.

[Figure]

Reply: Done.

Reply: Done.

Reply: Done.

Reply: Yes, we have the measurements during the sampling period and the measurement of the samples. This is a good point to be added to the manuscript. Consequently, we decided to add the following in the results section in page 7, line 2 as follows:

"Atmospheric water vapor concentration during the collection of samples had a mean value of 17 930 ± 369 ppm. This concentration changes with time during the different days of measurement (Figure 2). Between the measuring days 2 and 9 the water vapor concentration drops from 18000 ppm to less than 14000 ppm, while towards the measuring day 19 it increased with 1000 ppm more. This trend is tightly followed by the PVF sampling bags, followed by the LDPE with a larger difference and the MPE with small variations respect to the atmospheric water vapor concentration of the samples collection. This data shows that all the sampling bags exchange water vapor from and towards the atmosphere with a different degree of magnitude."

Page 7, line 26-27:

"… stable isotope measurements and water vapor concentrations, including the MPE bags. …"

[Figure]

Figure 2. Boxplots describing the concentration of atmospheric water vapor in ppm during the collection of samples and the posterior measurement of samples.

Comment 22. Page 7, Line 16: One single value is not very meaningful. Can you replicate this?
Reply: the authors are aware of the lack of representativity of only one sample. However, the design showed in page 6 lines 7-10 describe the cryogenic sampling. We were expecting to collect three samples for each pumping rate, however the device was not successful to collect enough sample with the lower pumping rate as it is mention in page 7, line 16-18. Despite this step back, we decided to add this sample to include all data collected. To replicate this sample it will be necessary to re-do all the experiment, however as the cryogenic samples collected with the pumping rate of 3 L min$^{-1}$ were suitable for analysis we do not see the need to replicate the  50 mL min$^{-1}$ samples. Also, this provides insights about the difficulties linked with low pumping rates for the collection of atmospheric water vapor.

Comment 23. Page 7, Line 24: How do you make sure that you trap "all" the conveyed air? What would be a measure for that?
Reply: if we measure the relative humidity of the air on the inlet and at the outlet of the cryogenic bath, we can have a measurement of the specific humidity of the air. Here, we can compare the difference between both flows and determine how efficient the cryogenic bath is. Consequently, we decided to add the following to the manuscript in Page 7, line 24:

"… from the conveyed air. However, as this is an open system that requires a constant flow of air and cannot be closed as the batch distillation (Koeniger *et al*, 2011; Vendramini and Sternberg, 2007), it requires to monitor the atmospheric water vapor concentration before and after the sample collection. This can be achieved measuring the air temperature and relative humidity in both, the inlet and outlet of the cryogenic bath. This information can be used to estimate the specific humidity of the air and evaluate the efficiency of the cryogenic bath. "

Comment 24. Page 7, Line 27: add ","
Reply: Done.

Comment 25. Page 7, Line 30: Again, how high/low were your values during the sampling?

Reply: in our experiment we did not measure the WVTR. The values showed on the manuscript are the reported ones for each material. However, following a previous recommendation of the reviewer (comment 21) we added the water vapor concentration during the sampling and posterior measurement of samples. That specific data showed the variation of the water vapor concentration among samples and measuring days. This, fulfill this request of information about how high/low these values were during sampling, meanwhile the WVTR is a characteristic that "defines the ability of a film to transfer water molecules depending on the relative humidity gradient" (page 7, lines 27-28 of this manuscript).

Comment 26. Page 8, Figure 2: A minimum of at least three samples is required to draw any conclusions.
Reply: see reply in comment 22.

Comment 27. Page 8, Line 9: I would not go so far, since you only have a limited amount of replicates.
Reply: we modify the sentence as follows:

"…sample collection. Atmospheric water vapor sampling with cryogenic baths provides suitable accuracy when the collection efficiency is high. However, this requires a suitable system to monitor the specific humidity of the air at the inlet and outlet of the cryogenic bath. However, there …"

Comment 28. Page 8, Line 11: Can you say something about material costs for the different methods/bags types? Can you say something about practicability of the cryo versus the bag method? Please provide some more suggestions to the reader in terms of selecting an appropriate method.
Reply:  Following this suggestion, we decided to add the following in page 7, after line 32:

"The suitability of every sampling method to collect atmospheric water vapor in the field will depend in their accuracy to keep unmodified the mass and isotope signature of every sample (Peters and Yakir, 2010). However, the logistics (e.g, the location, the travel time from and towards the laboratory, basic research infrastructure on the field) and the project budgets play an important role on the selection of the sampling methods. Assuming in this experiment the laboratory equipment (e.g, glassware, output cavity spectrometer, air temperature and relative humidity sensors) and logistics (e.g, all the expenses related to the travel costs to the sampling) as fixed costs, the main difference will rely on the selected sampling method. Carrying out this experiment in Europe in 2019, the MPE and PVF sampling bags have a cost of €12.0 per bag and the LDPE sampling bags of € 0.2 per bag with the disadvantage that each of them can be use only once. On the other hand, collecting 3 samples per hour with the cryogenic bath during 24 hours, the price per sample during one day is € 2. After thawing the sample and storing it properly within borosilicate vials, the sample can be store for longer periods of time. This method requires also the monitoring of the water vapor concentration during the sampling to have an indication of the collection efficiency."

Adding the following in the conclusions:

"MPE sampling bags are the more accurate and more expensive sampling method. The LDPE sampling bag is the cheapest sampling method with the limitation that the samples should be analyzed on the same day of collection. This method gives an additional restriction considering the transport time and susceptibility to exchange water vapor with the surrounding air. Finally, the cryogenic bath has an affordable price per sample if the project collects samples continuously, maximizes the use of the supplies such as the dry ice that tends, and monitor the collection efficiency with the estimation of the specific humidity of the sampled air"

Comment 29. Page 14, Line 5: "an" instead of "a"
Reply: Done.

**References**

Koeniger, P., Marshall, J. D., Link, T., and Mulch, A.: An inexpensive, fast, and reliable method for vacuum extraction of soil and plant water for stable isotope analyses by mass spectrometry, Rapid Communications in Mass Spectrometry, 25, 3041–3048,https://doi.org/10.1002/rcm.5198, 2011.

Peters, L. I. and Yakir, D.: A rapid method for the sampling of atmospheric water vapour for isotopic analysis, Rapid Communications in Mass Spectrometry, 24, 103–108, https://doi.org/10.1002/rcm.4359, 2010.

Vendramini, P. F. and Sternberg, L. d. S. L.: A faster plant stem-water extraction method, Rapid Communications in Mass Spectrometry, 21,164–168, https://doi.org/10.1002/rcm.2826, 2007.

---

## Referee Comment (RC3) · Anonymous Referee #3 · 20 Nov 2019

I am reviewing the manuscript of César Dionisio Jiménez–Rodríguez and colleagues, entitled "comparison of water vapor sampling techniques for stable isotope analysis" submitted as technical note and currently under discussion for HESS.

The authors investigated in the laboratory the short to mid-term (1-16 days) reliability of different plastic sampling bags for isotopic analysis of water vapor, as an affordable alternative to the online analysis with laser-based spectrometers. They highlight that only one commercial product (made out of MPE) provided reliable results, independently from storage time. The authors don't say clearly what could have been the cause of the change with time in isotopic composition of the sampled water vapor in the

two other types of bags, although they show the influence of the laboratory air during measurement. If the bags were not tight, then the influence of the laboratory air would not be restricted to the time of measurement, but would spread from sampling to measurement times (i.e., over the whole storage duration). To me, this points to a tightness problem during measurement. The laser spectrometer seemed to have sampled from a mix of laboratory air and bag air and measured some average-weighed water vapor isotopic composition. This should be thoroughly addressed by the authors.

I also urge the authors to remove the part on cryogenic extraction from the benchmarking, as it was not working well enough during the experiment and is off-topic.

I found the text to be difficult to read at times and generally not well structured. I am afraid it did not undergo sufficient internal review before it was submitted to HESS. I provide the authors with many comments and corrections below.

Technical comments:

Throughout the manuscript: - stick to past tense for things that happened in the past. - provide unit/dimension for all parameters and variables present in the different equations

Abstract P1L1. The first two sentences read awkward. I propose something like: "The isotopic characterization of water vapor samples can help describing evaporation processes. However the collection of water vapor, which historically involves phase change, may be associated with fractionation in case of incomplete sampling. "The isotope signature of a parcel of air was continuously monitored with a cavity..." P1L4-5. You did not sample the same air parcel with different methods simultaneously, rather you sampled from the same air parcel simultaneously with different methods.

1 Introduction

P1L11. The isotopes or isotopologues are the tracers, not their isotopic signatures.

P1L12. "Water stable isotopes..". Also move the deltas to the previous sentence. You

are not talking about their signatures here.

P1L14. Consider citing a more recent review on plant water use isotopic applications, e.g., Rothfuss and Javaux (2017).

P1L15. Remove "only" from this sentence: sampling precipitation conservatively is not that straightforward actually.

P1L17. Some sample cannot describe evaporation processes, please rephrase.

P1L18-19. "It is formed by water vapor originated from evapotranspiration (plant transpiration and soil evaporation) and of intercepted water..." What about free water surfaces (streams, lakes, etc.)?

P1L20-22. This should be shortened: go straight to keywords evapotranspiration partitioning.

P1L22. "has been..."?. It is still the case and will always be, right?

P2L4-15. This level of detail in this section is not needed. Also it is confusing: at this point, it is not clear anymore what you are referring to "water vapor"; is it the atmospheric water vapor or the evaporated water vapor? Since you are addressing the potential isotopic effects of the sampling bags during/following collection/storage of atmospheric water vapor, this first "technique" is off-topic.

P2L17. "collects"

P2L17-23. These two sentences are hard to read, should be merged and shortened, e.g. the "second method consists in..." (it is the operator who collects); "conveyed at..."; "requires sublimating...". What you need to say is that we have two different classes of measuring techniques: 1- online, direct and continuous and 2- offline. The second technique is then subdivided in two sub classes: a-direct (with sampling bags) and indirect (via sublimation).

P2L20-23. Again, these sentences are hard to read: a sample does not collect itself;

Interactive
comment

there is no such thing as "heavier isotope signatures". An isotopic signature does not "occur". You should write something like: "The collection of water vapor in cold-traps for isotopic analysis is only a reliable technique when recovery rate tends towards 1; an incomplete recovery of water vapor fractionates water stable isotopes following the Rayleigh distillation model"

P2L25-26. We measure the isotopic composition of water vapor, not the "water stable isotopes". Change "cavity output spectroscopy" to the general term "laser-based spectroscopy".

P2L31. What is a "storage unit"?

P3L1-3. I think there is a conceptual misunderstanding here: the equilibration bags for soil water vapor are water-tight (due to the hydrophobic nature of the plastic) but let the water vapor diffuse through their walls. The observed problems have not to do with "[liquid] water diffusing through the container wall"

P3L3-4. Why are you referring to "units" here? In the abstract you say "sampling bag". You should stick to one lexical field/terminology.

2 Methodology

2.1 Instrumentation and Measurements

P3L12. Why is mentioning the voltage of the water pump of interest to the reader? How do they relate to the "controlled water vapor concentration"? Please explain. Also you mean water vapor "mixing ratio", and not "concentration".

P3L13. What is a "sample set here"?

P3L17-18. "allows the connection to different sampling devices. . .". It is "inlet" or "port"? Stick to wording for ease of reading.

P3L18-21. Why do you need the altered water vapor source?

P3L23-24. "The first 3 minutes were discarded from the analysis to account for memory of the instrument to the previous sample". Users know that memory effect is due to the turnover rate of the volume ahead and inside the cell of the laser. No need to say this.

P3L24-25. Please revise grammar

Figure 1. I don't think the drawing is necessary. Text is enough to me. Also both liquid and vapor modes are missing the part of the analyzer itself. They should be renamed e.g. vapor and liquid "peripheral devices".

P4L1-5 & Eq. (1). This is also not needed. Also terminology is not correct: "hydrogen and oxygen isotopic ratios are expressed relatively to those of the SMOW" not "Isotope signatures (2H and 18O) [...] were expressed in respect to the Vienna Standard Mean Ocean Water (VSMOW)". A signature is the relative deviation of some ratio with respect to another, meaning it remains a ratio, only on another scale. So why is a "signature" termed as "relative concentration" when "ratio" isn't? They are all ratio of concentrations.

2.2 Water Vapor Correction

You address here the problem of the nonlinear response of the laser spectrometer to changes of water vapor mixing ratio, which translates into a dependency of the raw isotopic readings to water vapor mixing ratio in the measuring range. This section's text is overall not well structured. Choose your wording carefully and stick with it. It starts with the title: it should read something like "Water vapor isotopic calibration". The isotopic terminology is sometimes not adequate (e.g. "The raw signatures of $\delta$2H and $\delta$18O"). I advise you to be as precise as possible and write "water vapor mixing ratio" and not "concentration of water molecules" (does not apply here: the LGR measures water vapor mixing ratio, not concentration) or "ppm concentration" (does not mean anything).

P4L8-10. Revise sentence structure. It is hard to read.

P4L10-12. Revise sentence as well: how did you measure evaporated standard water at different water vapor mixing ratio? Is it by changing the water pump flow rate (hence the different voltage values mentioned earlier?) or by adapting the dry air flow rate? I understand you did a one-point calibration rather than a two-point (span) calibration? Have you investigated the laser dependency to water vapor mixing ratio at another value of isotopic composition?

Eqs (2-5): they yield to $\delta18O=\delta18Ostandard$ and $\delta2H=\delta2Hstandard$, so something does not add up here. Please revise. A "good" instrument is characterized by a slow time drift of a, b and c; is this the case?

2.3 Experimental Design

P5L15-16. "...the isotope signature of the stored air water vapor samples remains..." or the "the water vapor isotope signature of the stored air samples remains..."

Points1.-3. Avoid starting each sentence with "these"

P5L18. "According to the supplier,..."

P6L3. Until now, it wasn't clear that the experiment took place in the laboratory. It should be stated early on (abstract + listing of objectives in the introduction). Did you check for complete recovery for the fast (3 L min-1) option by, for, instance measuring the water vapor mixing ratio with the laser spectrometer at the outlet of the trap, or installing a second trap in series to observe if water vapor was collected in it?

P6L8. Was it a simple test tube or something more elaborated (i.e. with an inner collecting wall)? This needs further detail as this greatly impacts the ability of the trap to capture moisture.

2.4 Analysis

This section needs some streamlining: you are unnecessarily repeating yourself (P6L18: "The cross comparison was performed with the Z-score analysis" vs P6L28:

"The comparison was performed with the Zscore analysis").

P6L17. Revise sentence, e.g., "you compare the isotopic signature of the samples to that of the benchmark", not "to the benchmark".

P6L19. "(Orlowski et al., 2016; Wassenaar et al., 2012), where S is the isotope signature ($\delta$2H or $\delta$18O) of the bags air water vapor and cryogenic water samples, B is the benchmark water vapor isotope signature (WVIA), and $\mu$ is the target variability."

P6L25-26. Here, you are not testing the ability of the association trap+cryogenic bath to fully collect the air moisture, rather you are testing its reproducibility. See my previous comment on P6L3.

3 Results and Discussion

P7L2-3. Revise sentence ("the isotope signature of the benchmark . . . had an isotope signature")

P7L5-6. Please revise (edit grammar).

P7L15-24 and Fig. 2B. I don't understand the difference between "cryogenic bath" and "cryogenic test sample". There is no distinction made in the text here and before. . .but see my general comment: I don't see how cryogenic water extraction is relevant in your study. Also: use letter "d" for day, or "t" for time, but not "T" (stands for temperature usually).

Rothfuss, Y., and Javaux, M.: Reviews and syntheses: Isotopic approaches to quantify root water uptake and redistribution: a review and comparison of methods, Biogeosci. Disc., 14, 2199-2224, doi:10.5194/bg-14-2199-2017, 2017.

---

## Author Comment (AC4) · 4 Dec 2019

In blue we copied the comments of the reviewer, in black our replies.

I am reviewing the manuscript of César Dionisio Jiménez–Rodríguez and colleagues, entitled "comparison of water vapor sampling techniques for stable isotope analysis" submitted as technical note and currently under discussion for HESS. The authors investigated in the laboratory the short to mid-term (1-16 days) reliability of different plastic sampling bags for isotopic analysis of water vapor, as an affordable alternative to the online analysis with laser-based spectrometers. They highlight that only one commercial product (made out of MPE) provided reliable results, independently from storage time. The authors don't say clearly what could have been the cause of the change with time in isotopic composition of the sampled water vapor in the two other types of bags, although they show the influence of the laboratory air during measurement.

If the bags were not tight, then the influence of the laboratory air would not be restricted to the time of measurement, but would spread from sampling to measurement times (i.e., over the whole storage duration). To me, this points to a tightness problem during measurement. The laser spectrometer seemed to have sampled from a mix of laboratory air and bag air and measured some average-weighed water vapor isotopic composition. This should be thoroughly addressed by the authors.

Reply: there are three possibilities of exchange with the laboratory air:

1. Through the bag material
2. Through the valve during analysis
3. Through both, valve and material.

We think based on the results in figure 2 that #1 and #3 are minor in comparison to #2. Because MPE and PVF have the same valve system, where MPE is still performing well after 9 days whereas PVF follows the laboratory air. This is likely due to the higher Water Vapor Transmission Rate of the bag material.

So you are right that PVF is not only disturbed during analysis, but also during storage. However, it's surprising that PVF is seemly so much affected by the laboratory air during analysis. Since we did not measured the water vapor mixing ratio on all days we cannot concluded whether the air in the PVF bag is affected during storage and/or only during analysis. This should be investigated in further research. To clarify this point, we added the following paragraphs:

Following the reviewer's suggestion to elaborate more on this issue we proposed to add the following:

Page 7, Line 2:
"Water vapor mixing ratio during the collection of samples had a mean value of 17 930 ± 369 ppm. This concentration changes with time during the different days of measurement (Figure 2). Between the measuring days 2 and 9 the water vapor mixing ratio drops from 18000 ppm to less than 14000 ppm, while towards the measuring day 19 it increased with 1000 ppm more. This trend is tightly followed by the PVF sampling bags, followed by the LDPE with a larger difference and the MPE with small variations respect to the water vapor mixing ratio of the samples collection. This data shows that all the sampling bags exchange water vapor from and towards the atmosphere with a different degree of magnitude."

Page 8, Line 1:

"The tendency of drift towards the signature of the laboratory air could be linked to other factors such as welding quality between bag material and the valve (for MPE and PVF bags), fitting issues between the tubing connecting the sample bags to the MIU unit (all sample bags) or the inlet connection for the LDPE bags. In the case of MPE and PVF bags, the manufacturer states that the bags should not be filled more than 90% of their capacity. This practice could lead to the development of fissures between the air valve and the bag material that in the case of PVF bags due to their brittle properties respect to MPE of LDPE bags. An increment on the air pressure within the MPE bags can lead to the detachment of the air valve from the layers in which it is welded. LDPE bags are susceptible to leaking as a consequence of the inlet built with in-house materials that the presence of different joints can induce the filtering of the laboratory air. "

[Figure]

In new manuscript: Figure 2. Boxplots describing the water vapor mixing ration in ppm during the collection of samples and the posterior measurement of samples.

I also urge the authors to remove the part on cryogenic extraction from the benchmarking, as it was not working well enough during the experiment and is off-topic.
Reply:  the reviewer is right that the cryogenic bath during the collection of air with the sample bags was not carried out properly and gave some errors. However, the second time we did run the cryogenic bath and compared against the WVIA run simultaneously the data show the capacity of this sampling method. The second text provides information about another sampling procedure widely used for sampling air water vapor that requires and removing it will reduce the discussion about collection methods.

Aiming to skip this, we proposed to remove the cryogenic samples collected during the first test due to their lower performance and small number of samples. Also, we propose to keep the second test explained in page 6, lines 25-28 and the results from figure 2.B (submitted manuscript) as the proper evaluation of the cryogenic bath method against the WVIA. In this way, the analysis is still enriching with the evaluation of the cryogenic bath with a larger number of samples.

I found the text to be difficult to read at times and generally not well structured. I am afraid it did not undergo sufficient internal review before it was submitted to HESS. I provide the authors with many comments and corrections below.
Reply: the authors thank the reviewer for the suggestions to improve the manuscript.

Technical comments:
Throughout the manuscript: - stick to past tense for things that happened in the past. - provide unit/dimension for all parameters and variables present in the different equations
Reply: we surveyed and checked the manuscript to improve homogenized the verb tenses accordingly with the suggestion.

Abstract P1L1. The first two sentences read awkward. I propose something like:
"The isotopic characterization of water vapor samples can help describing evaporation processes. However the collection of water vapor, which historically involves phase change, may be associated with fractionation in case of incomplete sampling. "
Reply:  we improved the sentences accordingly.

"The isotope signature of a parcel of air was continuously monitored with a cavity. . ."
Reply: we improved accordingly.

 P1L4-5. You did not sample the same air parcel with different methods simultaneously, rather you sampled from the same air parcel simultaneously with different methods.
Reply: this sentence was improved as follows:

"… (benchmark). Sampling from the same air parcel simultaneously with 3 types of sampling bags made of …"

1 Introduction
P1L11. The isotopes or isotopologues are the tracers, not their isotopic signatures.
Reply: the δ was removed to refer only to the isotopologues. Also, we check the manuscript to fix this issue.

P1L12. "Water stable isotopes..". Also move the deltas to the previous sentence. You are not talking about their signatures here.
Reply: done.

P1L14. Consider citing a more recent review on plant water use isotopic applications,
e.g., Rothfuss and Javaux (2017).
Reply: thanks for the recommendation. This reference (Rothfuss and Javaux, 2017) was checked and included on the manuscript together with Schwendenmann *et al* (2015) and Wang *et al* (2017).

P1L15. Remove "only" from this sentence: sampling precipitation conservatively is not that straightforward actually.
Reply: done.

P1L17. Some sample cannot describe evaporation processes, please rephrase.
Reply:  this sentence was modified as follows:
"… samples. Water vapor samples such as transpired water or atmospheric water are good descriptors of the evaporation process , …"

P1L18-19. "It is formed by water vapor originated from evapotranspiration (plant transpiration and soil evaporation) and of intercepted water..." What about free water surfaces (streams, lakes, etc.)?
Reply: we include the free water surfaces to this definition and the respective reference as follows:

"… It is formed by water vapor originated from open water evaporation, plant transpiration, soil evaporation and the evaporation of intercepted water on wet surfaces (Abtew and Melesse, 2013; Savenije, 2004). …"

P1L20-22. This should be shortened: go straight to keywords evapotranspiration partitioning.
Reply: it was changed as follows:

"… Evaporation partitioning has been carried out …"

P1L22. "has been. . ."?. It is still the case and will always be, right?
Reply: verb tense was changed for present: "is"

P2L4-15. This level of detail in this section is not needed. Also it is confusing: at this point, it is not clear anymore what you are referring to "water vapor"; is it the atmospheric water vapor or the evaporated water vapor? Since you are addressing the potential isotopic effects of the sampling bags during/following collection/storage of atmospheric water vapor, this first "technique" is off-topic.
Reply: aiming to prevent this type of confusion, we simplified this paragraph as follows:

"Currently, the estimation of the isotope signatures of water vapor can be performed with three different methods. Firstly, the Craig-Gordon model (CG–model) (Craig and Gordon, 1965) determine the water vapor signature of evaporation originated from open waters (Horita et al., 2008) and has also been applied in transpiration and soil evaporation studies (Dubbert et al.,2013; Ferrio et al., 2009; Williams et al., 2004). The high sensitivity of $^{18}O$ to temperature makes some assumptions of this model unreliable for the application in soil evaporation or plant transpiration processes (Dubbert et al., 2013). The cryogenic bath is the second …"

P2L17. "collects"
Reply: done

P2L17-23. These two sentences are hard to read, should be merged and shortened, e.g. the "second method consists in. . ." (it is the operator who collects); "conveyed at. . ."; "requires sublimating...". What you need to say is that we have two different classes of measuring techniques: 1- online, direct and continuous and 2- offline. The second technique is then subdivided in two sub classes: a-direct (with sampling bags) and indirect (via sublimation).
Reply: see next reply.

P2L20-23. Again, these sentences are hard to read: a sample does not collect itself; there is no such thing as "heavier isotope signatures". An isotopic signature does not "occur". You should write something like: "The collection of water vapor in cold-traps for isotopic analysis is only a reliable technique when recovery rate tends towards 1; an incomplete recovery of water vapor fractionates water stable isotopes following the Rayleigh distillation model."
Reply: according with the suggestions from the reviewer, we improved these lines as follows:

"The second method consists in a cryogenic bath that allow the collection of atmospheric water vapor within a canister immerse in a cooling agent (e.g,liquid nitrogen) (IAEA, 2016; Kool et al., 2014), freezing the water vapor conveyed at a constant air flow into the collection canister (He and Smith, 1999; Sheppard, 1958; Wen et al., 2016). This sampling method for isotopic analysis is only a reliable technique when recovery rate tends towards 1 (Griffis, 2013). An incomplete recovery of water vapor fractionates the water stable isotopes following the Rayleigh distillation model (Kendall and Caldwell, 1998). The consequences of incomplete sample recoveries are similar to the effects as studied by Orlowski et al. (2018) for soil water extraction."

P2L25-26. We measure the isotopic composition of water vapor, not the "water stable isotopes".
Reply: this sentence was changed as follows:

"The most recent method involves direct measurements of the isotopic composition of water vapor using mass spectrometers or laser-based spectroscopy."

Change "cavity output spectroscopy" to the general term "laser-based spectroscopy".
Reply: we followed this recommendation and the manuscript was updated accordingly with the suggestion.

P2L31. What is a "storage unit"?
Reply: here we refer to the sampling bags.

P3L1-3. I think there is a conceptual misunderstanding here: the equilibration bags for soil water vapor are water-tight (due to the hydrophobic nature of the plastic) but let the water vapor diffuse through their walls. The observed problems have not to do with "[liquid] water diffusing through the container wall"
Reply: this sentence was fixed as follows:

"… Wassenaar et al., 2008). They underline the risk of water vapor diffusion through the wall container when using equilibrium bags of different materials to determine the soil water isotope signature. If their findings for equilibrium bags used in soil water measurements hold for air water vapor samples as well, is still unknown. The aim …"

P3L3-4. Why are you referring to "units" here? In the abstract you say "sampling bag". You should stick to one lexical field/terminology.
Reply: following the recommendation from the reviewers, we changed the term "storage units" for "sampling bags" in the whole document.

2 Methodology
2.1 Instrumentation and Measurements
P3L12. Why is mentioning the voltage of the water pump of interest to the reader?
How do they relate to the "controlled water vapor concentration"? Please explain.
Reply: the voltage pump is the only parameter that the user can adjust on the WVSS, because the built-in software controls the mixing water ratio by the power provided to the pump (Page 3, lines 12-13). Aiming to provide more details about this item, we followed the recommendation of Reviewer 1. Then we add Appendix C

Also you mean water vapor "mixing ratio", and not "concentration".

Reply: yes, it should be "water vapor mixing ratio". Aiming to homogenize the manuscript terminology, we checked and change this term accordingly.

Reply: we refer to every round that the MIU unit measures from the different inlets.

Reply: it is "inlet". We checked and fixed this through the whole document.

Reply: The idea behind adding an altered air source during the sampling is to be able to differentiate between samples during the laboratory procedure and post-processing of the data. This helped to carry out the data analysis because after the air passed by the drying element, the $^2$H and $^{18}$O signatures will change drastically. To clarify this, we proposed to improve the lines 18 and 19 of page 3 as follows:

"… post-processing of the data. The data obtained from this inlet was not used during the analysis as it was used only as a distinction mark between samples. The altered air …"

Reply: we modified the sentence and attached to the previous paragraph as follows:
"… with sampling intervals of 5 s. The first 3 minutes were discarded by the memory effect, calculating the average and standard deviation of each measurement based on the last 2 minutes of measurements per sample."

Reply: see previous reply.

Reply: the authors want to keep the diagram because it will help to people new in the topic to understand how the system is integrated. Also, we changed the diagram according to the reviewer's suggestion.

Reply: aiming to simplify this section to the reader, it was changed as follows:

"Stable water isotope signatures of air vapor and liquid samples were expressed in δ values (‰), representing the relative deviation from Vienna Standard Mean Ocean Water (VSMOW) (Craig, 1961)."

2.2 Water Vapor Correction
You address here the problem of the nonlinear response of the laser spectrometer to changes of water vapor mixing ratio, which translates into a dependency of the raw isotopic readings to water vapor mixing ratio in the measuring range. This section's text is overall not well structured. Choose your wording carefully and stick with it. It starts with the title: it should read something like "Water vapor isotopic calibration". The isotopic terminology is sometimes not adequate (e.g. "The raw signatures of δ2H and δ18O"). I advise you to be as precise as possible and write "water vapor mixing ratio" and not "concentration of water molecules" (does not apply here: the LGR measures water vapor mixing ratio, not concentration) or "ppm concentration" (does not mean anything).
Reply: this section was upgraded as follows:

"2.2  Water Vapor Isotopic Calibraton

Measurements of water vapor isotope signatures depend on the water vapor mixing ratio (ppm) and the specific drift of the laser spectrometer of the WIA unit, which makes it essential to correct each individual measurement (Aemisegger et al., 2012; Rambo et al., 2011; Kurita et al., 2012; Steen-Larsen et al., 2013, 2014). The drift of the used laser spectrometer was negligible, because the measurement period was not longer than six hours every day. In addition, the thermal control within the laser chamber provides stable measurements with a negligible drift as it is stated by the manufacturer (LGR, 2019). The calibration was performed with a standard water ($\delta^{18}O_{standard}$: -14.4, $\delta^2H_{standard}$:-104.9) injected into the WIA at different pumping rates depending on the pump voltage (see Section 2.1). The injection is controlled by a built–in software package that managed the WVISS pump and the DAS. This system allows the use of only one standard water to calibrate the isotope signatures carried out with the WVIA. The calibration procedure was performed every time the MIU start a new round of measurements (see Appendix D). The measured isotope signatures ($\delta^2H_{raw}$ and $\delta^{18}O_{raw}$) were calibrated using the correction factors ($\phi_O$ and $\phi_H$) determined based on the dependency of the isotope signatures of standard water ($\delta^2H_{standard}$ and $\delta^{18}O_{standard}$) to their water vapor mixing ratio ($w$) in ppm. The polynomial coefficients a, b and c in equations 2 and 3 were determined for every set of measurements per experiment (Rambo et al., 2011; Kurita et al., 2012; Steen-Larsen et al., 2013, 2014). Each data point used in equations 2 and 3 corresponds to the last minute of measurements for each voltage, obtaining an average based on 12 individual measurements for both stable isotopes and water vapor concentrations. The calibrated values of each stable isotope ($\delta^{18}O$ and $\delta^2H$) were determined with equations 4 and 5 (Rambo et al., 2011; Steen-Larsen et al., 2013)."

P4L8-10. Revise sentence structure. It is hard to read.
Reply: see previous reply about Section 2.2.

P4L10-12. Revise sentence as well: how did you measure evaporated standard water at different water vapor mixing ratio? Is it by changing the water pump flow rate (hence the different voltage values mentioned earlier?) or by adapting the dry air flow rate?
Reply:  see previous reply about Section 2.2.

I understand you did a one-point calibration rather than a two-point (span) calibration?
Reply:  yes, it is one point calibration due to the settings of the device. However, we followed the recommendations given by Rambo et al. (2011), Kurita et al. (2012) and Steen-Larsen et al. (2013) for this procedure (Page 5, Lines 4-8).

Have you investigated the laser dependency to water vapor mixing ratio at another value of isotopic composition?

Reply:  yes, we had checked with one additional isotopic composition ($\delta^{18}O$: -5.6 and $\delta^2H$: -40.8). The device follows the polynomial tendency as described by Rambo et al. (2011) but the coefficients a, b and c have to be constantly determined. Figure ## shows the results of that test.

[Figure]

Figure ##. Isotope signatures of the raw measurements (black boxplots) and calibrated values (blue boxplots) of the known water ($\delta^{18}O$: -5.6 and $\delta^2H$: -40.8).

Eqs (2-5): they yield to δ18O=δ18Ostandard and δ2H=δ2Hstandard, so something does not add up here. Please revise. A "good" instrument is characterized by a slow time drift of a, b and c; is this the case?

Reply:

The LGR analyser used during this experiment did not experience a significant drift during the measurements due to the little time running on a daily basis (less than six hours every day).  Additionally, the model of water isotope analyser (IWA) keeps a "negligible drift" as it is stated by the manufacturer (LGR, 2019).

To clarify this, we proposed to add the following sentence on Page 4, line 10:

"… Steen-Larsen et al., 2013, 2014). **The drift of the used laser spectrometer was negligible, because the measurement period was not longer than 6 hours every day. In addition, the thermal control within the laser chamber provides stable measurements with a negligible drift as it is stated by the manufacturer (LGR, 2019).** The correction of water vapor measurements was …"

Also, the authors agreed with the suggestion of one of the reviewers of adding an additional plot to the manuscript showing the relationship of the correction procedure applied. This plot shows the negligible drift of the laser in terms of the calibration factors $\phi_O$ and $\phi_H$. The plot will be added as an appendix to the manuscript as follows:

**Appendix C: Plots of the Water Vapor Correction Procedure**

[Figure]

**Figure C1.** Water vapor correction plots showing the variation of the water vapor mixing ratio (ppm) and the calibration factors $\phi_O$ and $\phi_H$. Plot A shows the variation of water vapor missing ratio in ppm for each voltage used on the WVISS pump during the correction procedure. Plots B and C show the polynomial relationships between the water vapor missing ratio and the calibration factors $\phi_O$ and $\phi_H$, respectively.

Also, equations 2 to 5 are uses to correct the raw isotope signatures given by the WVIA according with their dependency on the water vapor mixing ratios (Rambo et al, 2011). The following figure shows the calibration process during the water vapor sampling procedure of three hours. The grey areas correspond to the controlled water vapor mixing ratio of the known standard ($\delta^{18}O$: -14.4 and $\delta^{2}H$: -104.9). The polynomial relationship of the data set in between grey areas (equations 2 and 3) were used to correct the raw signatures of the sampled air based on the dependency of the water vapor mixing ratio. This calibration was done estimating the φ coefficient with the equations 2 and 3 for the raw measurements of the samples, applying later on equations 4 and 5 to retrieve the corrected values. As it is shown in the figure, this procedure does not change the raw measurements into the standard water. The coefficients a, b and c do not refer to slow time drift, they refer to the linear relationship showed by the mixing ratio and the raw isotope signature measured by the device. We proposed to add the following image as an Appendix to the manuscript to better understand the calibration process.

[Figure]

**2.3 Experimental Design**

P5L15-16. ". . .the isotope signature of the stored air water vapor samples remains. . ." or the "the water vapor isotope signature of the stored air samples remains. . ."
Reply: we changed it for:
"… the water vapor isotope signature of the stored air samples remains …"

However, this line was move up to improve the objectives section in the introduction following the recommendation of one reviewer.

**Points1.-3. Avoid starting each sentence with "these"**

Reply: each point was improved as follows:
"1. MPE: bags of 1L made of methalized polyethylene and manufactured with …
 2. PVF: bags of 1L made of …
 3. LDPE: bags of 1L made of …"

**P5L18. "According to the supplier,. . ."**
Reply: correction done.

P6L3. Until now, it wasn't clear that the experiment took place in the laboratory. It should be stated early on (abstract + listing of objectives in the introduction).
Reply: aiming to inform the reader earlier about the location of the sampling, we add the following:

Page 1, Line 3-4: "…The isotope signature of a parcel of air was continuously monitored in the laboratory with a laser-based spectroscopy …"

Also, the objectives in the introduction were improved according with the suggestions of all the reviewers as follows:

"The aim of this work is to evaluate different sampling procedures to collect atmospheric water vapor and analyze the stable water isotopes. This experiment tested whether the stored mass of water vapor sampled in the laboratory remained unchanged as well as whether the isotope signature of the stored water vapor remains consistent in time. We included three sampling bags to determine their suitability for sampling, storing and analyzing water vapor isotopes. The results were compared against a set of cryogenic samples and direct measurements performed with laser-based spectroscopy."

Did you check for complete recovery for the fast (3 L min-1) option by, for, instance measuring the water vapor mixing ratio with the laser spectrometer at the outlet of the trap, or installing a second trap in series to observe if water vapor was collected in it?
Reply: no, we did not evaluate the complete recovery of the sampled air.

P6L8. Was it a simple test tube or something more elaborated (i.e. with an inner collecting wall)? This needs further detail as this greatly impacts the ability of the trap to capture moisture.
Reply: following the suggestion of one of the reviewers, we add a diagram of the cryogenic bath used during this experiment as an appendix.

[Figure]

Figure C1. Cryogenic bath diagram for the collection of liquid samples of atmospheric water vapor.

2.4 Analysis
This section needs some streamlining: you are unnecessarily repeating yourself
(P6L18: "The cross comparison was performed with the Z-score analysis" vs P6L28: The comparison was performed with the Zscore analysis").
Reply: we removed the repeated sentence in Page 6, line 28.

P6L17. Revise sentence, e.g., "you compare the isotopic signature of the samples to that of the benchmark", not "to the benchmark".
Reply: we improved the sentence as follows:
"The consistency analysis of the isotopic signatures was performed comparing the isotope signatures of the samples against the Benchmark. …"

P6L19. "(Orlowski et al., 2016; Wassenaar et al., 2012), where S is the isotope signature ($\delta 2H$ or $\delta 18O$) of the bags air water vapor and cryogenic water samples, B is the benchmark water vapor isotope signature (WVIA), and µ is the target variability."
Reply: we add the correction to the sentence. Thanks!

3 Results and Discussion
P7L2-3. Revise sentence ("the isotope signature of the benchmark . . . had an isotope signature")
Reply: the sentence was improved as follows:

"The stable isotope signatures of the benchmark during the three hours of the experiment were -15.61±0.14‰ and -115.12±0.47‰ for $\delta^{18}O$ and $\delta^{2}H$, respectively. The benchmark represents the center point of both graphs in Figure …"

P7L5-6. Please revise (edit grammar).
Reply: the sentence was edited as follows:

"The isotope signature of the laboratory air water vapor was not constant on the different days when the measurements were performed."

Also: use letter "d" for day, or "t" for time, but not "T" (stands for temperature usually).
Reply: we modified Figure 2 using d instead of T when we refer to days as follows:

[Figure]

P6L25-26. Here, you are not testing the ability of the association trap+cryogenic bath to fully collect the air moisture, rather you are testing its reproducibility. See my previous comment on P6L3.

Reply: you are right. Here we are not testing the collection capacity, however the results from the second test shows the capacity of using the cryogenic bath as benchmark when the technique is carried out properly.

P7L15-24 and Fig. 2B. I don't understand the difference between "cryogenic bath" and "cryogenic test sample". There is no distinction made in the text here and before. . .but see my general comment: I don't see how cryogenic water extraction is relevant in your study.

Reply: there is no difference on the collection method between both samples. We choose different words to differentiate between the two samplings. Again, the authors strengthen the need to keep the second test where the cryogenic bath is tested carefully against the WVIA.

References

Abtew, W. and Melesse, A.: Wetland Evapotranspiration, pp. 93–108, Springer Netherlands, Dordrecht, https://doi.org/10.1007/978-94-007-4737-1_7, 2013.

Aemisegger, F., Sturm, P., Graf, P., Sodemann, H., Pfahl, S., Knohl, A., and Wernli, H.: Measuring variations of delta;18O and delta;2H inatmospheric water vapour using two commercial laser-based spectrometers: an instrument characterisation study, Atmospheric Measure-5ment Techniques, 5, 1491–1511, https://doi.org/10.5194/amt-5-1491-2012, 2012.

Craig, H.: Standard for Reporting Concentrations of Deuterium and Oxygen-18 in Natural Waters, Science, 133, 1833–1834,https://doi.org/10.1126/science.133.3467.1833, 1961.

Dubbert, M., Kübert, A., and Werner, C.: Impact of leaf traits on temporal dynamics of transpired oxygen isotope signatures and its impacton atmospheric vapor, Frontiers in plant science, 8, https://doi.org/10.3389/fpls.2017.00005, 2017.

Ferrio, J. P., Cuntz, M., Offermann, C., Siegwolf, R., Saurer, M., and Gessler, A.: Effect of water availability on leaf water iso-30topic enrichment in beech seedlings shows limitations of current fractionation models, Plant, Cell & Environment, 32, 1285–1296,https://doi.org/10.1111/j.1365-3040.2009.01996.x, 2009.

Griffis, T. J.: Tracing the flow of carbon dioxide and water vapor between the biosphere and atmosphere: A review of optical isotope tech-5niques and their application, Agricultural and Forest Meteorology, 174-175, 85 – 109, https://doi.org/10.1016/j.agrformet.2013.02.009,2013. 2013

He, H. and Smith, R. B.: Stable isotope composition of water vapor in the atmospheric boundary layer above the forests of New England,Journal of Geophysical Research: Atmospheres, 104, 11 657–11 673, https://doi.org/10.1029/1999JD900080, 1999.

Horita, J., Rozanski, K., and Cohen, S.: Isotope effects in the evaporation of water: a status report of the Craig–Gordon model, Isotopes in Environmental and Health Studies, 44, 23–49, https://doi.org/10.1080/10256010801887174, 2008.

IAEA: Supporting sampling and sample preparation tools for isotope and nuclear analysis. IAEA–TECDOC–1783, https://www.iaea.org/publications/10991/supporting-sampling-and-sample-preparation-tools-for-isotope-and-nuclear-analysis, 2016.

Kendall, C. and Caldwell, E. A.: Chapter 2 - Fundamentals of Isotope Geochemistry, in: Isotope Tracers in Catchment Hydrology, editedby KENDALL, C. and McDONNELL, J. J., pp. 51 – 86, Elsevier, Amsterdam, https://doi.org/10.1016/B978-0-444-81546-0.50009-4, http://www.sciencedirect.com/science/article/pii/B9780444815460500094, 1998.

Kool, D., Agam, N., Lazarovitch, N., Heitman, J., Sauer, T., and Ben-Gal, A.: A review of approaches for evapotranspiration partitioning,Agricultural and Forest Meteorology, 184, 56 – 70, https://doi.org/10.1016/j.agrformet.2013.09.003, 2014.

Orlowski, N., Breuer, L., Angeli, N., Boeckx, P., Brumbt, C., Cook, C. S., Dubbert, M., Dyckmans, J., Gallagher, B., Gralher, B., Herbstritt,B., Hervé-Fernández, P., Hissler, C., Koeniger, P., Legout, A., Macdonald, C. J., Oyarzún, C., Redelstein, R., Seidler, C., Siegwolf,R., Stumpp, C., Thomsen, S., Weiler, M., Werner, C., and McDonnell, J. J.: Inter-laboratory comparison of cryogenic water extraction systems for stable isotope analysis of soil water, Hydrology and Earth System Sciences, 22, 3619–3637, https://doi.org/10.5194/hess-22-153619-2018, 2018.

Rambo, J., Lai, C.-T., Farlin, J., Schroeder, M., and Bible, K.: On-Site Calibration for High Precision Measurements of Water Vapor IsotopeRatios Using Off-Axis Cavity-Enhanced Absorption Spectroscopy, Journal of Atmospheric and Oceanic Technology, 28, 1448–1457,https://doi.org/10.1175/JTECH-D-11-00053.1, 2011.

Rothfuss, Y., and Javaux, M.: Reviews and syntheses: Isotopic approaches to quantify root water uptake and redistribution: a review and comparison of methods, Biogeosci. Disc., 14, 2199-2224, doi:10.5194/bg-14-2199-2017, 2017.

Schwendenmann, L., Pendall, E., Sanchez-Bragado, R., Kunert, N., and Hölscher, D.: Tree water uptake in a tropical plantation varying in tree diversity: interspecific differences, seasonal shifts and complementarity, Ecohydrology, 8, 1–12, https://doi.org/10.1002/eco.1479,2015.

Sheppard, P.: Transfer across the earth's surface and through the air above, Quarterly Journal of the Royal Meteorological Society, 84,205–224, https://doi.org/10.1002/qj.49708436102, 1958.

Wang, J., Fu, B., Lu, N., and Zhang, L.: Seasonal variation in water uptake patterns of three plant species based on stable isotopes in thesemi-arid Loess Plateau, Science of The Total Environment, 609, 27 – 37, https://doi.org/10.1016/j.scitotenv.2017.07.133, 2017.

Wen, X., Yang, B., Sun, X., and Lee, X.: Evapotranspiration partitioning through in-situ oxygen isotope measurements in an oasis crop-land, Agricultural and Forest Meteorology, 230-231, 89 – 96, https://doi.org/https://doi.org/10.1016/j.agrformet.2015.12.003, oasis-desertsystem, 2016.

Williams, D., Cable, W., Hultine, K., Hoedjes, J., Yepez, E., Simonneaux, V., Er-Raki, S., Boulet, G., de Bruin, H., Chehbouni, A., Har-togensis, O., and Timouk, F.: Evapotranspiration components determined by stable isotope, sap flow and eddy covariance techniques,Agricultural and Forest Meteorology, 125, 241 – 258, https://doi.org/10.1016/j.agrformet.2004.04.008, 2004.